# Comparison of Orthognathic Surgery Outcomes Between Patients With and Without Underlying High-Risk Conditions: A Multidisciplinary Team-Based Approach and Practical Guidelines

**DOI:** 10.3390/jcm8111760

**Published:** 2019-10-23

**Authors:** Pang-Yun Chou, Rafael Denadai, Chit Chen, Betty Chien-Jung Pai, Kai-Hsiang Hsu, Che-Tzu Chang, Dax Pascasio, Jennifer Ann-Jou Lin, Yu-Ray Chen, Lun-Jou Lo

**Affiliations:** 1Department of Plastic and Reconstructive Surgery and Craniofacial Research Center, Chang Gung Memorial Hospital, Chang Gung University, Taoyuan 333, Taiwan; chou.asapulu@gmail.com (P.-Y.C.); denadai.rafael@hotmail.com (R.D.); daxpascasio@gmail.com (D.P.); jenniferannlin@yahoo.com.tw (J.A.-J.L.); uraychen@cgmh.org.tw (Y.-R.C.); 2Department of Anesthesia, Chang Gung Memorial Hospital, Taoyuan 333, Taiwan; cacpan@cgmh.org.tw; 3Department of Craniofacial Orthodontics, Chang Gung Memorial Hospital, Taoyuan 333, Taiwan; pai0072@cgmh.org.tw; 4Division of Neonatology, Department of Pediatrics, Chang Gung Memorial Hospital, Taoyuan 333, Taiwan, Chang Gung University, Taoyuan 333, Taiwan; khsu@cgmh.org.tw; 5Division of Rheumatology, Allergy, and Immunology, Chang Gung Memorial Hospital, Taoyuan 333, Taiwan; chang3109@cgmh.org.tw

**Keywords:** orthognathic surgery, underlying conditions, safety, perioperative care, multidisciplinary care, guidelines

## Abstract

Orthognathic surgery (OGS) has been successfully adopted for managing a wide spectrum of skeletofacial deformities, but patients with underlying conditions have not been treated using OGS because of the relatively high risk of surgical anesthetic procedure-related complications. This study compared the OGS outcomes of patients with and without underlying high-risk conditions, which were managed using a comprehensive, multidisciplinary team-based OGS approach with condition-specific practical perioperative care guidelines. Data of surgical anesthetic outcomes (intraoperative blood loss, operative duration, need for prolonged intubation, reintubation, admission to an intensive care unit, length of hospital stay, and complications), facial esthetic outcomes (professional panel assessment), and patient-reported outcomes (FACE-Q social function, psychological well-being, and satisfaction with decision scales) of consecutive patients with underlying high-risk conditions (*n =* 30) treated between 2004 and 2017 were retrospectively collected. Patients without these underlying conditions (*n =* 30) treated during the same period were randomly selected for comparison. FACE-Q reports of 50 ethnicity-, sex-, and age-matched healthy individuals were obtained. The OGS-treated patients with and without underlying high-risk conditions differed significantly in their American Society of Anesthesiologists Physical Status (ASA-PS) classification (*p* < 0.05), Charlson comorbidity scores, and Elixhauser comorbidity scores. The two groups presented similar outcomes (all *p* > 0.05) for all assessed outcome parameters, except for intraoperative blood loss (*p* < 0.001; 974.3 ± 592.7 mL vs. 657.6 ± 355.0 mL). Comparisons with healthy individuals revealed no significant differences (*p* > 0.05). The patients with underlying high-risk conditions treated using a multidisciplinary team-based OGS approach and the patients without the conditions had similar OGS-related outcomes.

## 1. Introduction

Facial deformities associated with malocclusion cause functional and esthetic impairment and adversely affect quality of life, body image perception, self-esteem, and social interaction [1,2]. Therefore, multidisciplinary team-based surgical approaches have been considered to be crucial for providing not only reconstruction of facial deformities but also support for improving patients’ psychosocial conditions [1,2].

Orthognathic surgery (OGS) may be used to correct abnormal skeletal relationships between the maxilla and mandible in various congenital and acquired facial and occlusal deformities [2,3,4,5,6]. OGS is a technically demanding procedure performed on facial bones, and it is associated with prolonged general anesthesia and substantial blood loss [2,3,4,5,6]. Therefore, subgroups of patients with underlying conditions or chronic underlying diseases in addition to the principal diagnosis of facial or occlusal abnormalities have not been treated using OGS because of the relatively high risk of complications.

In contrast to life-saving surgical interventions, OGS is considered a life-enhancing intervention that focuses on not only correcting dental occlusion and improving facial esthetics but also improving the quality of life and psychosocial condition of patients [1,2]. Therefore, patients with facial deformity, malocclusion, and underlying high-risk conditions can be considered potentially eligible candidates for receiving potential OGS-related benefits. However, this particular subset of patients may require additional care and management methods along with the conventional OGS approach. For an elective life-enhancing procedure to be considered ethically acceptable, the surgical-related risks must be weighed against its anticipated benefits, particularly in patients affected by medical factors that may potentially increase perioperative complications [7,8].

In our referral craniofacial center for OGS treatment, we have treated a high volume of patients in the past 30 years [9,10,11,12,13,14,15,16,17,18]. Our evolving comprehensive OGS care method, which is based on a multidisciplinary team-based OGS approach with condition-specific practical perioperative guidelines, has permitted us to successfully maximize the risk–benefit ratio of OGS in patients with underlying high-risk conditions. Although several OGS-related outcomes have previously been assessed [9,10,11,12,13,14,15,16,17,18], the management and outcomes of this specific subgroup of patients with underlying conditions have yet to be formally addressed.

The purposes of the present study were to assess the OGS treatment-related outcomes in patients with underlying high-risk conditions, to compare these outcomes with those in patients without underlying conditions, and to describe the current multidisciplinary team-based OGS approach. We hypothesized that the patients with underlying high-risk conditions would present a higher rate of OGS-related complications than patients with no underlying high-risk conditions.

## 2. Methods

This retrospective observational study was conducted on consecutive patients with underlying high-risk conditions (high-risk group) who had undergone OGS treatment for correction of facial deformities and skeletal Class II or III malocclusion between 2004 and 2017 in a single craniofacial center (Figure 1). Patients with no underlying conditions (regular-risk group) who had undergone OGS treatment between 2004 and 2017 were randomly selected for comparative analysis. Demographic (age and sex), clinical (diagnosis of underlying conditions, type of OGS procedure, American Society of Anesthesiologists Physical Status [ASA-PS] classification, Charlson comorbidity score, and Elixhauser comorbidity score) [19,20,21], and outcome data (surgical anesthetic-related outcome, facial esthetic-related outcome, and patient-reported outcome) were collected through digital medical records and interviews with all included patients.

The study was approved by an institutional review board (Chang Gung Medical Foundation, 201700659B0) and complied with the 1975 Declaration of Helsinki, as amended in 1983.

### 2.1. Multidisciplinary Approach

A comprehensive, multidisciplinary team-based OGS approach has been used in our craniofacial center (Figure 2 and Figure 3; See Appendix A). A member of the anesthetic team has been acting as the coordinator of this approach, which has enabled the streamlining of the perioperative processes, including screening, referral, optimization, and application of condition-specific practical guidelines. Both surgical and anesthetic teams are responsible for initial screening processes for underlying conditions. In this study, patients in the regular- or low-risk group (conditions such as including controlled hypertension, type II diabetes mellitus, obesity with body mass index < 40 kg/m^2^, and smoking habit) were managed according to the regular ASA recommendations for elective surgery [19]. However, patients with underlying conditions that were considered to be of high risk for OGS underwent additional evaluations by medical specialists, regardless of the strict definitions provided by the ASA criteria.

In general, in our center, a team of social workers actively helps in maintaining the agility and regularity of schedules for multidisciplinary visits, and a team of psychosociologists prepares patients for the perioperative process. A shared decision-making process between patients, their families, and our team has also been adopted for setting realistic expectations about OGS results, limitations of certain dentofacial changes, expected risks and complications, and postoperative recovery.

### 2.2. Practical Guidelines for Perioperative Care

The condition-specific medical specialty approach has been divided into three steps: assessment, optimization, and planning. For preoperative assessment, clinical evaluation, generic or specific (if available) grading systems for each condition, and complementary tests have been used to detect presumed conditions and to assess the severity of already known conditions. Based on a patient’s risk profile, a preoperative optimization is performed. Our specialist team provides condition-specific guideline plans with practical recommendations for perioperative care. Currently, we have guidelines for genetic, autoimmune, endocrine, neurocutaneous, infectious, psychiatric, and congenital heart disorders based on an evolving 14-year experience in managing patients with rare and common conditions that are considered to be of high risk for OGS (See Appendix A). New guidelines are constantly introduced if new clinical diagnoses are encountered by our team. If the same diagnosis is observed again, the existing guidelines are updated accordingly. For updating the guidelines, sequential criteria have been applied to each condition, including previously published condition-specific perioperative recommendations, the best available evidence, or available evidence.

### 2.3. Outcome Assessment

This study assessed three outcome parameters, including surgical anesthetic-related outcome, facial esthetic-related outcome, and patient-reported outcome. We assessed surgical anesthetic outcomes (such as intraoperative blood loss, operative duration, need for extended nasotracheal intubation after final wound closure, need for reintubation, requirement of admission or transfer to an intensive care unit, requirement of medication to support blood pressure after surgery, length of hospital stay, and need for readmission within 30 days after discharge for any reason), procedure-related complications (such as wound dehiscence, wound infection, postoperative hemorrhage/hematoma, fibrous union, nonunion, undesired fractures, and necrosis of bone segments), and systemic complications (such as venous thromboembolic disease and disorders of the ocular, cardiovascular, respiratory, urinary, and nervous systems). Need for revision surgery was defined as any revisionary bone and/or soft tissue procedure requested or required to improve occlusal, maxillary, mandible, and/or chin morphology within the follow-up period.

For assessing facial esthetic outcomes, an external blinded panel of seven plastic surgeons (comprising four men; 3–20 years after board certification) randomly rated the patients’ preoperative and 12-month postoperative standardized frontal and profile photographic views (Microsoft PowerPoint for Mac, Microsoft Corporation, USA and 15-inch MacBook Pro, Apple, Inc., USA) by using a previously published 7-point Likert scale for facial esthetic assessment on the basis of the following parameters: beautiful, attractive, and pleasant [22]. Moreover, 10% of the photographs were randomly selected and repeated during the assessment to determine intra-rater reliability.

For collecting data on patient-reported outcomes, the validated Mandarin Chinese version of the FACE-Q questionnaire [1,23,24] was administered to the included OGS-treated patients in the postoperative period (>12 months). Three specific scales were employed: the social function, psychological well-being, and satisfaction with decision scales. The sum of the scores for each scale was converted to an equivalent Rash score ranging from 0 to 100, with higher values indicating a better outcome. FACE-Q data from 50 healthy Taiwanese Chinese individuals (no history of craniofacial deformity, trauma, or surgery) were retrieved from the Chang Gung Craniofacial Research Center database, adjusted for matching factors (age and sex), and used for comparative analysis.

### 2.4. Statistical Analysis

Descriptive statistical data are summarized as mean ± standard deviation. The data distribution was verified using the Kolmogorov–Smirnov test, and the t test, Kruskal–Wallis test, and Wilcoxon signed-rank test were performed accordingly. The derived intra- and inter-rater reliability scores were excellent (intraclass correlation coefficient = 0.90 to 0.94) for the facial esthetic parameters. Two-sided *p* values of <0.05 were considered statistically significant. All analyses were performed using IBM SPSS software version 20.0 (IBM Corp., Armonk, NY, USA).

## 3. Results

This study included 60 OGS-treated patients in the high-risk (*n* = 30, See Appendix A) and regular-risk (*n* = 30) groups. The two groups of patients had similar distributions of age, sex, and type of OGS parameters (single-splint two-jaw surgery for correcting facial deformity associated with malocclusion). However, the groups differed significantly (*p* < 0.05) in their ASA-PS classification scores. In the high-risk group, the average Charlson comorbidity and Elixhauser comorbidity scores were 1.5 ± 1.5 and 2.9 ± 4.7, respectively; the types of comorbidities observed in the high-risk group were not observed in the regular-risk group (Table 1).

The high- and regular-risk groups did not differ significantly in all surgical anesthetic outcomes, facial esthetic outcomes, and patient-reported outcome parameters, except for intraoperative blood loss. The high-risk group exhibited a significantly (*p* < 0.05) higher volume of blood loss than the regular-risk group (Table 1). Comparisons of the OGS-treated patients with the matched healthy individuals (social function = 70.8 ± 23.1 and psychological well-being = 73.7 ± 22.4) revealed no significant differences in FACE-Q report scores.

All OGS-treated patients were extubated before transfer to the recovery room care unit, and no patient required admission or transfer to the intensive care unit, reintubation, postoperative medication to manage blood pressure, or readmission within 30 days after discharge. Systemic complications, procedure-related complications, or the need for revisionary surgery was not reported in this cohort (Figure 4 and Figure 5).

## 4. Discussion

In this comparative retrospective study, we showed similar results in high- and regular-risk groups for all tested parameters, i.e., surgical anesthetic-related outcome, facial aesthetic-related outcome, and patient-reported outcome, with an exception for the volume of blood loss parameter. Moreover, high- and regular-risk groups of OGS-treated patients presented a similar patient-reported outcome to matched healthy individuals.

The perioperative effects of underlying conditions have been acknowledged in multiple studies [25,26,27,28]. The anesthetic surgical literature reports outcome studies conducted on patients with underlying conditions who underwent surgery and were managed using a structured and multidisciplinary approach [25,26,27]; however, these studies have not formally described OGS procedures. Some studies on OGS have included the average ASA-PS or Charlson comorbidity scores but have not reported the type of underlying conditions, whereas other OGS studies describing the diagnosis of underlying conditions have primarily included patients with conditions (smoking, hypertension, and diabetes) that are commonly observed in all surgical contexts [4,5,29,30,31,32].

To expand on these previous reports, the present study compared the OGS-related outcomes of high- and regular-risk groups that were managed by an OGS-specific multidisciplinary approach in the past 14 years. Most of the underlying conditions considered in this study have not been previously reported in the context of OGS procedures, and only a few conditions have been described in some case reports [33,34,35]. As some of these underlying conditions are common worldwide, it is probable that the corresponding patients may not have received OGS due to the potential risk of complications or the treatment providers may not have specified details of high-risk conditions in their publications.

When treating patients with underlying conditions, our multidisciplinary team was observed to not regard them as a homogeneous group because each diagnosis has specific nuances. The results of this study show that the high-risk group had higher ASA-PS, Charlson, and Elixhauser scores than did the regular-risk group. These metrics exhibited significant differences, which are suitable for scientific comparisons [36,37,38,39,40]. However, consistent definitions for high-risk patients or conditions or a gold standard risk scale have yet to be developed [19,20,21,25,27,36,37,38,39,40,41,42,43]. Existing validated risk indices rely on fixed variables that do not necessarily record the nature, severity, and perception of the specific conditions in each patient receiving a particular surgical procedure [19,20,21,25,27,36,37,38,39,40,41,42,43]. In the ASA-PS classification system, a patient classified as ASA II is defined as a low-risk patient [19]. However, some of our patients who were classified as ASA II were considered to be of high risk for OGS procedures because their underlying conditions presented numerous challenges for anesthetic administration, surgery, and follow-up.

Some aspects of OGS procedures should be meticulously considered for each possible underlying condition encountered by the treating team, including hypotensive anesthesia in long operative durations with substantial blood loss (e.g., additional risk in patients with systemic lupus erythematosus, patients receiving hormone therapy (transgender patients), and patients with endocrinological and heart conditions due to potential systemic decompensation, deep vein thrombosis, and pulmonary embolism), intraoral incisions and several bone osteotomy lines with large bone mobilizations (e.g., patients with autoimmune diseases, receiving corticosteroid treatment, with human immunodeficiency virus infection, and with osteogenesis imperfecta due to potential infections, bone segmental necrosis, fibrous union, nonunion, and undesired fractures), and the need for regular postoperative appointments for orthodontic adjustments (e.g., patients with underlying psychiatric conditions who may have body dysmorphic problems or not adhere to follow-up).

Some aspects of OGS described as relevant to perioperative care in OGS may also be crucial for other types of surgical interventions; however, facial surgical procedures that may match OGS in terms of its unique aspects and requirements for preoperative diagnosis and planning for long-term targets are currently unavailable. Therefore, instead of using or adapting a protocol established for another elective facial surgical procedure, we have developed an OGS-specific model. Our comprehensive, multidisciplinary team-based OGS approach has been established based on the OGS principles that have been developed in our center over the past decades [9,10,11,12,13,14,15,16,17,18] and merged published principles for perioperative management (e.g., multidisciplinary team, perioperative personalized care matrix, integrated care pathway, perioperative risk optimization and management planning tool, and patient-centered care) [26,27,44,45].

In our center, the combination of orthodontic (surgery-first approach with postoperatively accelerated orthodontic tooth movement) [9,10,17,18], surgical (single-splint two-jaw surgery technique with modified face bow) [11,12,13,14], and anesthetic (tranexamic acid, hypotensive anesthesia, regional blocks, local anesthesia, and autologous blood transfusion) [15,16] principles and techniques with need-based three-dimensional advanced technology (preoperative simulation and intraoperative translation of virtual surgery with patient-specific printed surgical guides and digital occlusion set up and real time navigation) [11,12,13,14] has positively affected the planning, execution, and outcomes of OGS.

These anesthetic-, orthodontic-, and surgical-based interventions are customized according to the needs of each patient and have reduced the overall treatment time, rate of OGS-related complications, and need for revision surgery during follow-up. Moreover, unlike postoperative OGS care in other centers, we have not adopted intermaxillary fixation, nasogastric tube insertion, or routine admission for intensive care unit patients, thus improving patient and parent comfort and well-being after surgery and reducing the overall treatment costs [9,10,11,12,13,14,15,16,17,18]. Using this approach, we have described reproducible, stable, and consistent outcomes for vertical, transverse, and anteroposterior facial dimensions and occlusion-related parameters in patients with open bite, skeletal Class II and III deformities, and challenging facial asymmetric deformities [9,10,11,12,13,14,15,16,17,18]. 

In addition to these technical strategies, other factors have been considered to be key in determining the success of OGS in patients with underlying high-risk conditions. A multidisciplinary and proactive preoperative assessment is highly beneficial for detecting possible underlying conditions and simultaneously enabling early preoperative interventions and intra- and post-operative planning. Medical specialists constitute an integral part of our approach because they provide timely diagnosis, risk assessment, treatment decisions, and perioperative pathway planning, including condition-specific preoperative optimization. Moreover, the logistic format of our craniofacial center within our quaternary hospital system has facilitated the timing from first consultation to OGS while avoiding unnecessary delays in treatment, such as waiting lists for visiting subspecialists or complementary examination.

Although our OGS approach is based on the principles of patient-centered care, we provide a list of disease-specific practical guidelines. The shared decision-making process in our approach has enabled patients and their parents to understand the surgical process and the associated risks. Increasing patients’ engagement with the perioperative process is crucial to ensure their adherence to orthodontic and medical recommendations and to postsurgical care. The positive FACE-Q psychosocial outcomes observed in patients with underlying high-risk conditions probably resulted from the rigorous preoperative selection processes and the education of patients who are deemed to be stable, motivated, and compliant by a multidisciplinary team, including a psychologist and psychiatrist.

Studies that have examined outcomes across various types of surgical procedures have demonstrated that patients with a high comorbidity burden are more prone to postoperative complications, such as wound healing problems, infections, and organ- or system-related problems (i.e., problems in the vascular, urinary, digestive, and nervous systems) than those without comorbidities [25,26,27,28]. From the existing OGS-related data, patients with relatively high ASA-PS and Charlson scores or a high comorbidity burden have also been determined to be prone to complications, an extended length of hospital stay, and high medical costs [4,5,29,30,31,32]. However, differences in study design and data collection impair a reliable head-to-head comparison between our results and those of the mentioned previous studies [4,5,29,30,31,32]. Using our comprehensive OGS-specific approach, we demonstrated that achieving similar outcomes in high- and regular-risk groups is possible, with OGS-treated patients exhibiting similar FACE-Q reports to matched healthy individuals. All measured parameters (anesthetic surgical outcomes, procedure-based outcomes, and patient-reported outcomes) were considered key targets of OGS procedures [2]. Achieving this balance between risks and benefits is paramount to continue the use of this approach in our center.

Despite the similar outcomes, on average, the high-risk group had a higher volume of blood loss than the regular-risk group. Based on the modified Johns Hopkins surgical severity criteria [46], two-jaw OGS may be considered as a moderately to highly invasive procedure, with a potential blood loss of >1500 mL. In addition to general factors (presence or absence of underlying conditions) that may explain the differences in blood loss in our study, other OGS-related aspects (types of bone osteotomies and mobilizations) have been associated with the volume of blood loss [6,15,16,47]. Multidisciplinary teams should anticipate substantial blood loss during OGS procedures in patients with underlying high-risk conditions, and future investigations should appropriately assess the predictors of intraoperative bleeding in this specific cohort of patients.

According to the findings of the present study, we can recommend that patients with underlying high-risk conditions can safely undergo OGS procedures and subsequently achieve crucial endpoints. However, we do not recommend that this particular cohort of patients receive the conventional OGS approach. The current OGS-related outcomes should be interpreted as being obtained by a national reference hospital [9,10,11,12,13,14,15,16,17,18] for OGS; the hospital is considered a high-volume center with surgeons and anesthesiologists who manage a high volume of patients [29,30,31] by using a comprehensive multidisciplinary team-based OGS approach (integration of technical expertise with advanced technologies) associated with condition-specific guidelines for perioperative care. An increase in operative volume has been associated with benefits in reducing complications for patients undergoing various surgical procedures, including OGS [29,30,31,48]. The regionalization of all OGS patients to high-volume centers is an ongoing debate [29,30,31], but based on this study and the rationale from previous surgical experiences [29,30,31,48,49], patients with high-risk conditions should receive OGS procedures in structured, specialized centers. Our described OGS-specific model may be adopted in centers without established algorithms or protocols to treat medically complex patients. This should be an adaptive and dynamic process because our approach may be enhanced and adapted to each center by modifying or introducing orthodontic, surgical, and anesthetic principles and adding condition-specific guidelines. Relevant clinical features for activation and coordination of the masticatory muscles, i.e., the function generating bite principle, the well-designed functional appliance, and the periodontal mechanoreceptor concept [50,51,52,53], should also be considered in future management approaches to enhance the delivery of patient-specific OGS care.

The limitations of this study include an inherent bias associated with retrospective designs. Our sample was primarily comprised of young adult patients, which also reflects our OGS practice [9,10,11,12,13,14,15,16,17,18]. Therefore, we did not determine outcomes in older patients, a growing subgroup of patients receiving OGS treatment. We assessed three main endpoints in OGS, but some relevant parameters were not measured, including the total cost-effectiveness feature. We included different types of underlying high-risk conditions, but the complete spectrum of potential conditions was not addressed. Furthermore, we did not stratify the patients by disease-specific grade of severity. Although the adopted measures (ASA-PS, Charlson, and Elixhauser scores) have been extensively validated, their limitations should be considered [19,20,21,25,27,36,37,38,39,40,41]. We did not use the Surgical Outcome Risk Tool or ACS NSQIP surgical risk calculator [42,43] because these two tools do not allow input of multiple CPT codes, and the CPT code for two-jaw surgery is unavailable. Modifying existing measures or creating an OGS-specific risk calculator may improve outcome studies.

Despite these limitations, the present study is useful for health care professionals, health policymakers, and patients by providing the outcomes of an OGS-specific multidisciplinary model matched to patients’ needs. Other centers managing a specific subgroup of challenging patients with high-risk conditions should also publish their particular approaches and OGS-related outcomes. The published results will provide the basis to expand the criteria for OGS in the future and will support the spread of this possibility of treatment to referral health care providers (general dentists and orthodontists) [54], anesthesiologists, ENT, head and neck surgeons, plastic surgeons, maxillofacial surgeons, oral surgeons, and patients and their parents.

## 5. Conclusions

This study demonstrated an efficient OGS care delivery model for patients with different underlying high-risk conditions by using a comprehensive, multidisciplinary OGS team-based approach with condition-specific practical guidelines.

## Figures and Tables

**Figure 1 jcm-08-01760-f001:**
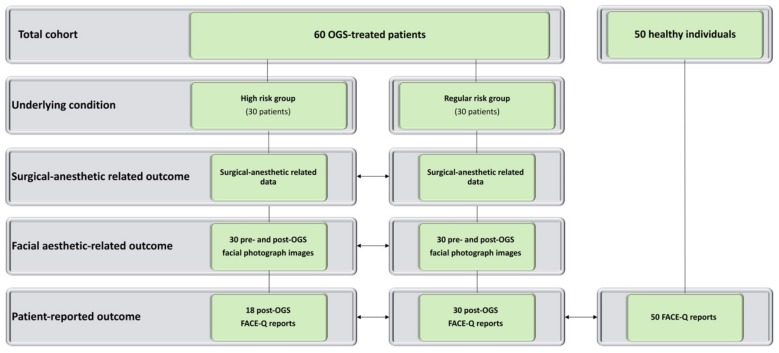
Flowchart for inclusion of orthognathic surgery (OGS)-treated patients with (high-risk patients) and without (regular-risk) underlying high-risk conditions.

**Figure 2 jcm-08-01760-f002:**
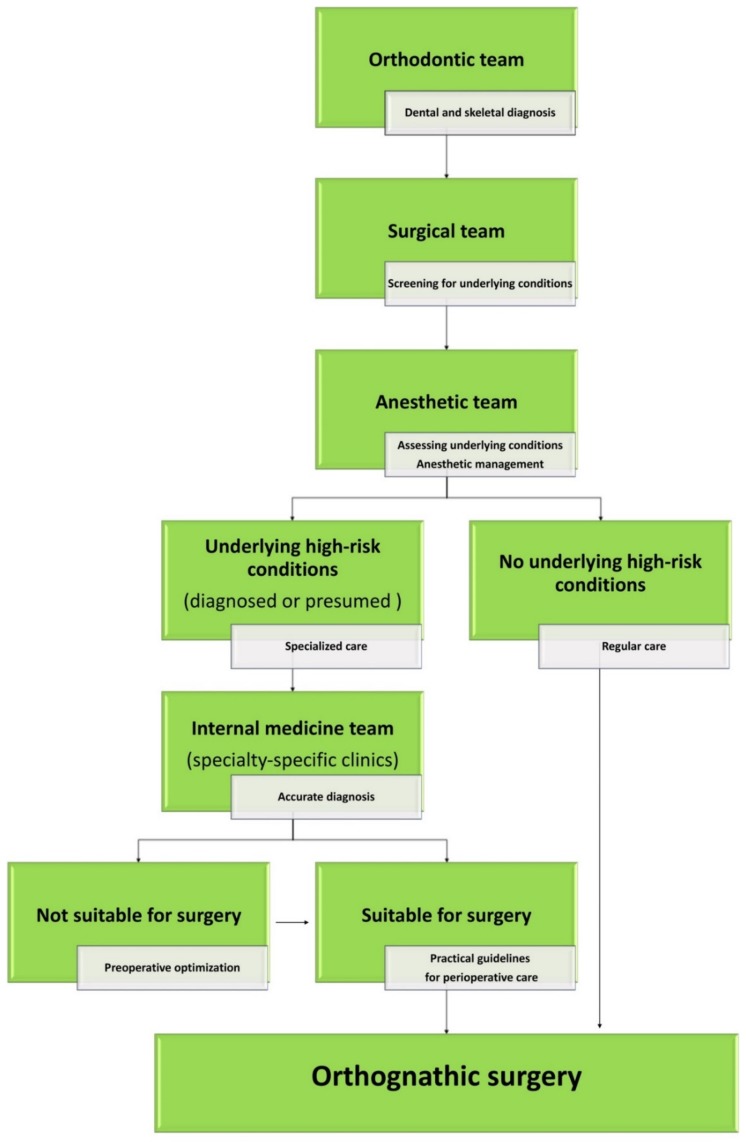
Multidisciplinary orthognathic surgery team-based approach. For patients without underlying high-risk conditions (regular-risk group), preoperative assessment (general and surgically focused history, physical examinations, and recommendations based on the American Society of Anesthesiologists Physical Status (ASA-PS) scores) conducted by the surgical and anesthetic teams are typically sufficient. For patients receiving diagnosed or presumed underlying high-risk conditions (high-risk group), specialty-specific consultations are required. Based on an accurate diagnosis (including complementary tests), patients may have their orthognathic surgery programmed or a preoperative optimization may be required before surgery. Condition-specific guidelines with practical recommendation for perioperative care are provided by the medical specialist team.

**Figure 3 jcm-08-01760-f003:**
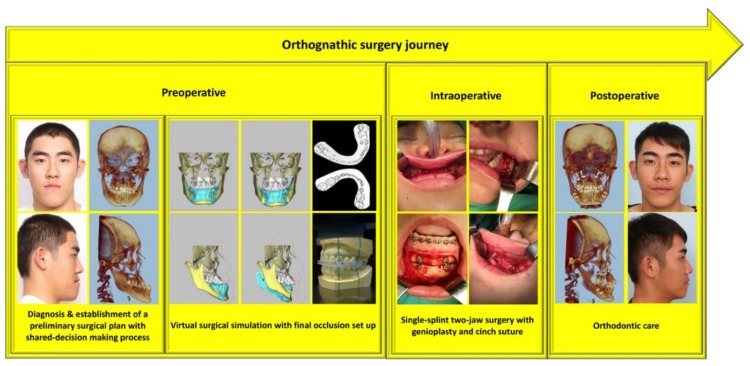
Multistep orthognathic surgery journey from the initial preoperative multidisciplinary-based consultation to reintegration into society after correction of facial deformity and malocclusion. Patient provided written consent for the use of his image.

**Figure 4 jcm-08-01760-f004:**
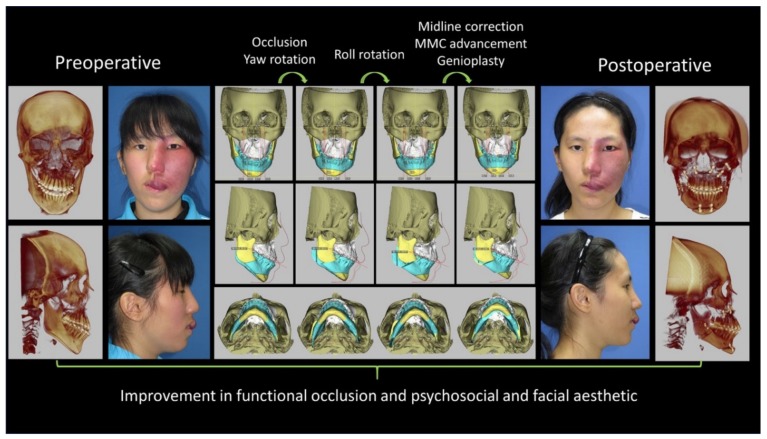
Case of a 16-year-old girl with Sturge Weber syndrome with a left face vascular malformation. The involvement of skeletal and soft tissues was extensive. She had received several sclerosing agent injections and laser treatments to reduce the vascular lesions. She experienced an episode of prolonged bleeding after a tooth extraction. Her facial deformity was characterized by left facial overgrowth and malocclusion with anterior open bite. Computed tomography angiography showed localized hypervascularity but no intracranial vascular malformation. Three-dimensional simulation was performed for surgical planning. She received Le Fort I, bilateral sagittal split of the ramus, genioplasty, and reduction of the left zygoma and buccal fat pad. Intraoperative blood oozing was noted, but was under control. Postoperative intermaxillary immobilization was not required, and the course was uneventful. One year later, she received the second stage of soft tissue reduction. The patient and her parents were satisfied with the treatment. (MMC: maxillomandibular complex). Patient provided written consent for the use of her image.

**Figure 5 jcm-08-01760-f005:**
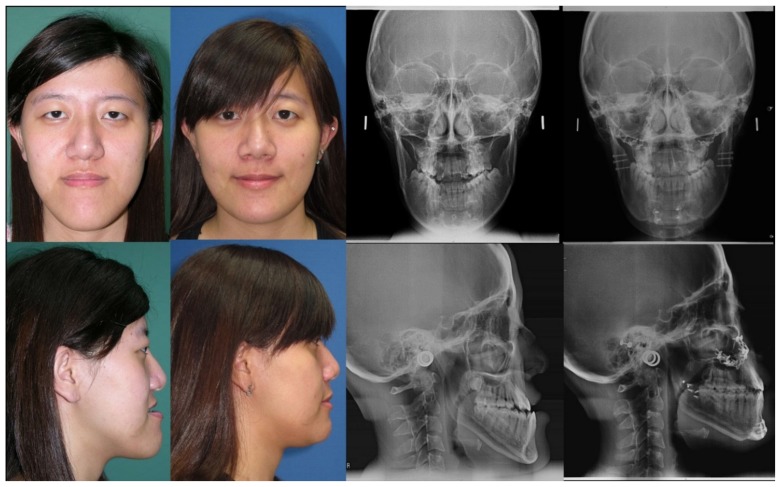
Case of an 18-year-old woman with a facial deformity characterized by a long face, mandibular protrusion, and chin deviation. She had a history of operation for biliary atresia during infancy, and liver cirrhosis and esophageal varices during adulthood. Her underlying conditions were adequately controlled. She underwent orthognathic surgery with Le Fort I, Wassmund osteotomy, bilateral sagittal split of the ramus, and genioplasty, with no postoperative intermaxillary immobilization. Her postoperative course was uneventful, and the results on occlusion and facial appearance were satisfactory. At follow-up 3 years after surgery, she appeared satisfied and reported psychosocial improvement. Patient provided written consent for the use of her image.

**Table 1 jcm-08-01760-t001:** Characteristics of and comparative analyses for orthognathic surgery-treated patients with and without underlying high-risk conditions.

Parameters	Orthognathic Surgery-Treated Patients		*p*
	High Risk Group (*n* = 30)	Regular Risk Group (*n* = 30)	
**Gender n (%)**			
Male/Female	10 (33.3)/20 (66.7)	12 (40)/18 (60)	>0.05
Age (years, m ± sd)	25.7 ± 7.4	26.3 ± 4.2	>0.05
Type of underlying disease n (%)			
Genetic	8 (26.6)		
Autoimmune	8 (26.6)		
Endocrine	5 (16.7)		
Neuro-cutaneous	3 (10)		
Infection	2 (6.7)		
Congenital cardiac	2 (6.7)		
Psychiatric	2 (6.7)		
ASA n (%)			
I/II/III	3 (10)/23 (76.7)/4 (13.3)	30 (100)/0 (0)/0 (0)	<0.001
Charlson comorbidity score m ± sd	1.5 ± 1.5 (1–6)	0 ± 0	
Number of Charlson comorbidities *n* (%)	11 (36.6)	0 (0)	
Charlson score = 1 *n* (%)	9 (30)		
Charlson score = 2 *n* (%)	1 (3.3)		
Charlson score = 6 *n* (%)	1 (3.3)		
Elixhauser comorbidity score m ± sd	2.9 ± 4.7 (−3–11)	0 ± 0	
Number of Elixhauser comorbidities n (%)	20 (66.7)	0 (0)	
Elixhauser score = −1 to −7 n (%)	4 (13.3)		
Elixhauser score = 0 *n* (%)	5 (16.8)		
Elixhauser score = 1 to 5 *n* (%)	7 (23.3)		
Elixhauser score = 6+ (%)	4 (13.3)		
Surgical anesthetic outcomes (m ± sd)			
Blood loss (mL, m ± sd)	974.3 ± 592.7	657.6 ± 355.0	<0.001
Operation time (min, m ± sd)	344.5 ± 106.2	347 ± 83.9	>0.05
Hospital length of stay (days, m ± sd)	4.0 ± 0.8	3.7 ± 0.8	>0.05
Facial aesthetic outcomes (m ± sd)			
Aesthetic (pre/post)	2.2 ± 1.1/5.1 ± 1.5	2.5 ± 1.0/5.3 ± 1.5	>0.05
Attractive (pre/post)	2.2 ± 0.9/4.6 ± 1.2	2.4 ± 0.9/4.9 ± 1.3	>0.05
Pleasant (pre/post)	2.0 ± 1.0/4.5 ± 1.6	2.2 ± 1.1/4.8 ± 1.5	>0.05
Post-OGS FACE-Q outcomes (m ± sd)			
Social function	71.5 ± 16.6	69.8 ± 22.1	>0.05
Psychological well being	72.3 ± 23.2	70.47 ± 22.3	>0.05
Satisfaction with decision	77.3 ± 14.6	74.7 ± 23.4	>0.05

m, mean; sd, standard deviation; pre, preoperative; post, postoperative; OGS, orthognathic surgery; ASA-PS, American Society of Anesthesiologists Physical Status Classification System; –, not applicable; Note: The proportions of binary comorbidity variables available for Charlson index (17 conditions) differed from those available for Elixhauser index (29 conditions). The Charlson weights assigned to each comorbidity ranged from +1 to +6, whereas the Elixhauser weights ranged from −7 to +12. For Elixhauser scores, a negative weight does not denote diseases as preventive factors because it is an artifact resulting from the original statistical methodology.

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
