# Peer review of "Comparison of Orthognathic Surgery Outcomes Between Patients With and Without Underlying High-Risk Conditions: A Multidisciplinary Team-Based Approach and Practical Guidelines"

_jcm, 2019, doi:10.3390/jcm8111760_

Round 1
Reviewer 1 Report
In the manuscript entitled: “Comparison of Orthognathic Surgery Outcomes between Patients with and without Underlying High- Risk Conditions: A Multidisciplinary Team-Based Approach and Practical Guidelines” the authors compared the OGS outcomes of patients with and without underlying high-risk conditions, which were managed using a comprehensive, multidisciplinary team-based OGS approach with condition-specific practical perioperative care guidelines.
The authors found that OGS-treated patients with and without underlying high-risk conditions differed significantly in their American Society of Anesthesiologists Physical Status (ASA-PS) classification, Charlson comorbidity scores, and Elixhauser comorbidity scores. The two groups presented similar outcomes for all assessed outcome parameters, except for intraoperative blood loss. Comparisons with normal individuals revealed no significant differences.
The authors concluded that patients with underlying high-risk conditions treated using a multidisciplinary team- based OGS approach and the patients without the conditions had similar OGS-related outcomes.
Major comments:
In general, the idea and innovation of this study, regards the analysis of orthognathic surgery outcomes 2 between patients with and without underlying high-3 risk conditions is interesting, because the role of these outcomes are validated but further studies on this topic could be an innovative issue in this field could be open an innovative matter of debate in literature by adding new information. Moreover, there are few reports in the literature that studied this interesting topic with this kind of study design.
The study was well conducted by the authors; However, there are some concerns to revise that are described below.
The introduction section resumes the existing knowledge regarding facial deformities.
However, as the importance of the topic, the reviewer strongly recommends to update the literature through read, discuss and cites in the references with great attention all of those recent interesting articles, that helps the authors to better introduce and discuss the aim of the study in light of the some factors related with the orthodontic treatment prior the surgery and on the periodontal mechanoreceptors involved during chewing and relative orthognathic surgery: 1) Isola G, Ramaglia L, Cordasco G, Lucchese A, Fiorillo L, Matarese G. The effect of a functional appliance in the management of temporomandibular joint disorders in patients with juvenile idiopathic arthritis. Minerva Stomatol. 2017 Feb;66(1):1-8. 2) Piancino MG, Isola G, Cannavale R, Cutroneo G, Vermiglio G, Bracco P, Anastasi GP. From periodontal mechanoreceptors to chewing motor control: A systematic review. Arch Oral Biol. 2017 Jun;78:109-121.
The authors should be better specified, at the end of the introduction section, the rational of the study and the relative null-hypothesis. In the material and methods section, should better clarify the outcome assessment and the FACE-Q questionnaire.
The discussion section appears well organized with the relevant paper that support the conclusions, even if the authors should better discuss the relationship between chewing dysfunction and orthognathic surgery. The conclusion should reinforce in light of the discussions.
In conclusion, I am sure that the authors are fine clinicians who achieve very nice results with their adopted protocol. However, this study, in my view does not in its current form satisfy a very high scientific requirement for publication in this journal and requests a revision before publication.
Minor Comments:
Abstract:
Better formulate the introduction section by better describe the background
Introduction:
Page 2, Line 59: please add the relative sentence
Discussion
Please add a specific sentence that clarifies the results obtained in the first part of the discussion Page 10 last paragraph of discussion: Please reorganize this paragraph that is not clear
Reviewer 2 Report
This article presents an extensive report on a multidisciplinary protocol for the management of patients with facial deformities. The authors present and discuss their protocol ("multidisciplinary team-based OGS approach with condition-specific practical perioperative guidelines") in a clear and concise manner. Furthermore, the authors assess high-risk patient outcomes and compare them to the "normal-risk" cohort. Overall, a well-drafted and written manuscript that is informative for a wide range of medical specialties that deal with craniofacial anomalies. The supplementary data is outstanding and highly informative.
Suggestions:
Figure 1
Suggest changing "50 normal individuals" to "50 healthy individuals". The term "normal" is rather subjective.
Figures 3 and 4 and 5. Did the individuals that are clearly visible consent to the publication of their pictures? Please provide adequate proof of their consent.
Round 2
Reviewer 1 Report
In the R1 version of the manuscript entitled: “Comparison of Orthognathic Surgery Outcomes between Patients with and without Underlying High-Risk Conditions: A Multidisciplinary Team-Based Approach and Practical Guidelines” the authors followed all the issues suggested by the reviewer. Though the changes based on the reviewer comments, almost of the criticisms were carefully analysed and solved.
I have carefully evaluated all parts of the manuscript.